

# On Parameter Bias in Earthquake Sequence Models using Data Assimilation

Arundhuti Banerjee[1], Ylona van Dinther[2], and Femke C. Vossepoel[1]

[1]Department of Geoscience and Engineering, Delft University of Technology, Stevinweg 1, 2628 CN Delft, the Netherlands.
[2]Department of Earth Sciences, Utrecht University, Princetonlaan 4,3584 CB Utrecht, the Netherlands.

**Correspondence:** Femke C. Vossepoel ([F.C.Vossepoel@tudelft.nl])

**Abstract.** The feasibility of physics-based forecasting of earthquakes depends on how well models can be calibrated to represent earthquake scenarios given uncertainties in both models and data. We investigate whether data assimilation can estimate current and future fault states, i.e., slip rate and shear stress, in the presence of a bias in the friction parameter. We perform state estimation as well as combined state-parameter estimation using a sequential importance resampling particle filter in a 0D
generalization of the Burridge–Knopoff spring-block model with rate-and-state friction. Minor changes in the friction parameter $\epsilon$ can lead to different state trajectories and earthquake characteristics. The performance of data assimilation in estimating the fault state in the presence of a parameter bias in $\epsilon$ depends on the magnitude of the bias. A small parameter bias in $\epsilon$ (+3 %) can be compensated very well using state estimation ($R^2 = 0.99$), whereas an intermediate bias ($-14$ %) can only be compensated partly ($R^2 = 0.47$). When increasing particle spread by accounting for model error and an additional resampling
step $R^2$ increases to 0.61. However, when there is a large bias ($-43$ %) in $\epsilon$, only state-parameter estimation can fully account for the parameter bias ($R^2 = 0.97$). Simultaneous state- and parameter estimation thus effectively separates error contributions from friction and shear stress to correctly estimate current and future shear stress and slip rate. This illustrates the potential of data assimilation for estimation of earthquake sequences and provides insight into its application in other non-linear processes with uncertain parameters.

## 1 Introduction

Earthquake hazard quantification requires estimates and uncertainties of e.g., long-term average recurrence rate of earthquakes. Hence modeling of earthquake sequences may help to understand and forecast the processes that determine these recurrence intervals. Physics-based models of the fault are therefore needed to predict the time when a subsequent earthquake will occur (Barbot et al. (2012)). Using these models, we calculate fault velocity and stresses for the entire earthquake sequence by solving
the quasi-dynamic equation of motion with laboratory-derived rate and state dependent friction laws. Most of these physics-based models of seismicity are designed to reproduce general characteristics of earthquakes. To tune and synchronize the model to observed reality Barbot et al. (2012), data assimilation can be useful. Data assimilation combines prior information from simulations of a physics-based model with information in the form of observations to obtain the best possible description of a dynamical system and its uncertainty, (e.g., Evensen et al., 2022). While data assimilation originates from weather forecasting



and oceanography (e.g., Daley, 1997; Bertino et al., 2003; Kalnay, 2003; Vossepoel and van Leeuwen, 2007), few studies have
introduced data assimilation for the purpose of earthquake forecasting (e.g., Van Dinther et al., 2019; Werner et al., 2011;
Hirahara and Nishikiori, 2019; Hori et al., 2014; Llenos and McGuire, 2011).

Applying data assimilation methods using physics-based models to predict the occurrence time of earthquakes is a highly
challenging task, because e.g., (i) the governing equations in these physics-based-models may not be sufficiently accurate to
forecast the earthquake cycles, (ii) earthquakes generally occur on faults located at depths of several to tens of kilometers and
we typically need to rely on indirect and noisy measurements to observe the conditions of the fault, and (iii) the state variables
and parameters in the models are highly uncertain and state variables may have multi-modal distributions. Several studies
have considered uncertainties in parameters using data assimilation in earthquake-cycle models (e.g. Kano et al., 2010, 2013;
Fakuda et al., 2009; Werner et al., 2011). Van Dinther et al. (2019) used a state estimation method to quantify uncertainty in the
state variables using a numerical model of an analogue subduction zone. Frictional parameters are important in the evolution
of fault slip as they largely determine the recurrence interval of earthquakes. Poorly known or misrepresented parameters can
thus introduce a bias, which can be an important source of uncertainty in the model. If this bias is not corrected, the forecasts
obtained using the forward model can be misleading. In previous studies, either the frictional parameters have been estimated as
part of the data assimilation, or assumed to be perfectly known. In this study, we will investigate the ability of state-estimation
methods to correctly update the states and compensate for a parameter bias and we will investigate the ability of state-parameter
estimation methods to reduce or even remove it.

The objective of this paper is to evaluate the effectiveness of data assimilation for state estimation and state-parameter
estimation in the presence of a parameter bias. To address this, we consider various cases: one set of cases where the parameter
is assumed to be known but has a biased value, and one set of cases where we the parameter is estimated along with the state. In
these cases, we model fault slip across faults seperating tectonic plates using a spring-slider model, which is assumed to obey
a rate-and-state-dependent friction formulation (e.g. Ruina, 1983; Erickson et al., 2008, 2011; Dieterich, 1979). We assimilate
observations of fault-slip velocity and fault shear stress using a particle filter to estimate the fault states.

## 2   Methods

### 2.1   Data assimilation framework

Let the vector $\mathbf{x}_t$ be the state of a model describing a dynamic system at time index $t = 1, \ldots, T$. We assume that the state is
evolving from time $t - 1$ to $t$ according to

$$\mathbf{x_t} = M_{t-1,t}(\mathbf{x_{t-1}}, \boldsymbol{\xi}) + \eta_{\mathbf{t}}, \tag{1}$$

where $\boldsymbol{\xi}$ represents a vector containing the model parameters and $M_{(t-1,t)}$ is the nonlinear operator describing the time-
evolution of the system. Here, $t$ is the time step for model integration. Acknowledging that the model is not perfect, we use $\boldsymbol{\eta}_t$
to represent the model error.





Consider a vector $\mathbf{y}$ that contains all observations $y_{t_o}$ at time $t_o = 1, \ldots, T_o$ as $\mathbf{y} = (y_1, y_2, \ldots, y_{t_o}, \ldots, y_{T_o})$. These observations are taken the 'true state' $\tilde{\mathbf{x}}_t$ for each time $t_o$, which is not necessarily coinciding with the model time steps. Observation $y_{t_o}$ can be related to the true state at that moment as follows

$$\mathbf{y}_{t_o} = H(\tilde{\mathbf{x}_{t_o}}) + \boldsymbol{\beta}_{t_o}, \tag{2}$$

where $H$ is the observational operator that maps the model to the data and $\boldsymbol{\beta}_{t_o}$ is the observation noise error. We assume observational errors to be independent and uncorrelated. Using these definitions of $\mathbf{x}$ and $\mathbf{y}$, we apply Bayes' theorem to obtain the posterior distribution of the state estimate

$$p(\mathbf{x} \mid \mathbf{y}) = \frac{p(\mathbf{y} \mid \mathbf{x})p(\mathbf{x})}{p(\mathbf{y})}, \tag{3}$$

where $p(\mathbf{y} \mid \mathbf{x})$ is the likelihood, $p(\mathbf{x})$ is the prior density and $p(\mathbf{y})$ the marginal density of the observations that can be
considered a normalization factor.

### 2.1.1 Particle filter

In the following, we describe a data-assimilation method for a generic state vector $\mathbf{z}$, which can represent the state $\mathbf{x}$ of the system, i.e., the variables that change over time, the parameters $\boldsymbol{\xi}$ of the system, which we assume to remain constant, or both. In case of state estimation, the state vector is $\mathbf{z} = \mathbf{x}$ and in case of state- and parameter estimation, the state vector also includes
the parameters $\mathbf{z} = [\mathbf{x}, \boldsymbol{\xi}]^{\mathrm{T}}$.

A Monte-Carlo representation in the form of particles can be used to approximate the posterior PDF $p(\mathbf{z}|\mathbf{y})$ at observation time $t_o$ as

$$p(\mathbf{z}_{0:t_o}|\mathbf{y}_{1:t_o}) = \frac{1}{N} \sum_{i=1}^{N} \delta(\mathbf{z}_{0:t_o} - \mathbf{z}_{0:t_o,j}), \tag{4}$$

where $N$ is the number of Monte-Carlo realizations, with $i$ the counter of these realizations.
We refer to these realizations as particles, while in other studies they may be referred to as ensemble members.

When we insert Eq.4 in Eq.3, and replace $\mathbf{x}$ by $\mathbf{z}$, we obtain an expression for the posterior distribution of the state vector $\mathbf{z}$

$$p(\mathbf{z}_{0:t_o}|\mathbf{y}_{1:t_o}) = \frac{1}{N} \sum_{i=1}^{N} w_i \delta(\mathbf{z}_{0:t_o} - \mathbf{z}_{0:t_o,j}). \tag{5}$$

Here the weights $w_i$ for each particle $i$ are

$$w_i = \frac{p(\mathbf{y}_{1:t_o,i} \mid \mathbf{z}_{0:t_o,i})}{\frac{1}{N} \sum_{i=1}^{N} p(\mathbf{y}_{1:t_o,i} \mid \mathbf{z}_{0:t_o,i})}, \tag{6}$$

where $p(\mathbf{y}_{1:t_o,i} \mid \mathbf{z}_{0:t_o,i})$ is the likelihood belonging to the observations in the time period 1 to $t_o$. As the observations $\mathbf{y}$ are fixed, this likelihood is a function of the vector $\mathbf{z}$ for each particle $i$, as discussed in Van Leeuwen (2015). Assuming that the





model operator describes a Markov process and the state at a certain (assimilation) time is completely determined by the model operator acting on the state of the (assimilation) time before, we write the weights as

$$w_i = \frac{p(\mathbf{y}_{t_o,i} \mid \mathbf{z}_{t_o,i})}{\frac{1}{N} \sum_{i=1}^{N} p(\mathbf{y}_{t_o,i} \mid \mathbf{z}_{t_o,i})}. \tag{7}$$

The values of the weights are thus determined by the likelihood $p(\mathbf{y}_{t_o}|\mathbf{z}_{t_o})$. Particles with a higher weight are closer to the observation than the ones with a lower weight. For the derivation of this method see Van Leeuwen (2009).

As the number of particles is typically too small to have sufficient samples of the prior, we observe that, as time progresses, most of the particles obtain negligible weights, whereas one or few particles obtain a very high weight. This is commonly referred to as filter degeneracy (e.g., Snyder et al. (2008)).

In the present study, the likelihood $p(y|\mathbf{z}_t)$ that defines the likelihood at a particular time $t$ for observation $y_{t_o}$ is assumed to be given by the Lorentz function

$$p(y_{t_o} \mid \mathbf{z}_{t_o}) = \frac{1}{1 + \frac{[y_{t_o} - H(\mathbf{z}_{t_o})]^2}{\beta_{t_o}^2}}, \tag{8}$$

where $\beta_{t_o}^2$ is the variance of the observation noise and $\mathbf{z}_{t_o}$ is the state vector at time $t_o$. The Lorentz function is chosen instead of a Gaussian function as the wider tails leads to less filter degeneracy (Vossepoel and van Leeuwen, 2007).

To further avoid degeneracy, we introduce a resampling step known as sequential importance resampling (SIR). This resampling discards particles with very low weights, while duplicating particles with high weight. Resampling is performed if the effective sample size $N_{eff}$ exceeds a threshold value. The effective sample size is given as

$$N_{eff} = \frac{1}{\sum w_i^2}. \tag{9}$$

and the threshold is typically chosen to be half the particle size, i.e. $N/2$.

In our study we have implemented a systematic resampling algorithm as it provides better estimates as compared to other resampling methods used in data assimilation [Hol et al. (2006)]. For a description of this and other resampling techniques, see, e.g., Doucet et al. (2001).

## 2.2 Forward model

### 2.2.1 Spring-block slider with rate-and-state friction

A simplified, computationally efficient description of earthquake sequences is provided by a spring-block slider system often referred to as the Burridge-Knopoff (BK) model for frictional sliding (Burridge and Knopoff, 1967). Such 0D and 1D models have been shown to retain key features of periodic sequences on homogeneous faults, including quantitative estimates of its recurrence interval and maximum coseismic slip upon using the calculated stress rate from 2/3D models (Li et al., 2021). It simplifies stick-slip motion of a fault caused by the adjacent movement of two tectonic plates using an assembly of springs and blocks (Fig. 1a). The blocks are connected by springs with spring constant $k_c$ and pulled by a plate moving with a uniform





velocity $V$ via another set of springs having spring constants $k_p$. The blocks represent one side of the fault, where the fault line is the contact surface between the blocks and the rough surface the blocks are placed upon. The equation of motion of the $i^{th}$ block in a 1-D chain of blocks of equal mass m is described by (Cartwright et al., 1997, Eq. 1)

$$m\ddot{z}_i = k_c(z_{i+1} - 2z_i + z_{i-1}) - k_p(z_i - Vt) - F_i(\dot{z}_i), \tag{10}$$

where $z_i$ is the position of block $i$, or the displacement from its initial position, and $F_i(\dot{z}_i)$ describes the frictional force.

We describe the frictional force using a laboratory-derived rate-and-state dependent friction formulation (e.g., Dieterich, 1979; Ruina, 1983; Marone, 1998). This empirical formulation has been used successfully over the last decades to describe the dynamics of sequences of earthquakes, including its spontaneous nucleation, propagation, arrest and postseismic slip in response to tectonic loading (e.g., Lapusta et al., 2000; Lapusta and Barbot, 2012). As in Rice and Tse (1986), the equation of

motion (Eq. 10) for rate-and-state dependent friction equations assuming a slip law for the evolution of state variable $\theta$ (Ruina, 1983) can be written as (Erickson et al., 2011, Eq. 6)

$$F_i(\dot{z}_i, \Theta) = \sigma_n \left( f^* + \Theta_i + a \ln \frac{\dot{z}_i}{v_0} \right), \tag{11}$$

$$\dot{\Theta}_i = -\left( \frac{\dot{z}_i}{L} \right) \left( \Theta_i + b \ln \frac{\dot{z}_i}{v_0} \right). \tag{12}$$

Here the capital notation $\Theta$ refers to $b \cdot \ln \frac{v_0 \theta}{L}$, which is an equivalent yet more convenient description for the rate-and-state friction state variable $\theta$, which can be interpreted as the change in interface strength from reference friction $f^*$ (Nakatani, 2001). Furthermore, $\sigma_n$ is normal stress, $v_0$ is the reference velocity assumed to be equal to the plate velocity $V$ below, and $a$, $b$ and $L$ are the associated rate-and-state friction parameters. After a sudden change in slip rate, parameter $a$ states the direct change of friction. Parameter $b$ describes the evolution towards a new steady-state friction coefficient, where characteristic slip

distance $L$ describes the distance taken by state variable $\theta$ to reach a new steady-state $\theta$.

In this study, we consider a zero-dimensional (0D) version analyzing a single spring-block slider (Fig. 1a). This model does not take into account the spatio-temporal correlation between different blocks or fault segments. Eq. 10 can be rewritten for a single block for non-dimensional slip of the block with respect to an initial position on the driving plate $u$ ($z$ in Fig. 1) and non-dimensional slip rate $v$ and solved by assuming $v_p = v_0$. This can be simplified into three partial differential equations

written in terms of dimensionless variables (using $\theta = A\hat{\theta}$, $v = v_0\hat{v}$, $u = L\hat{u}$ and $t = \frac{L}{v_0}\hat{t}$, where ˆ indicates the dimensional variables) as (updated from Erickson et al., 2008, Eq. 2)

$$\dot{\Theta} = -v(\Theta + (1 + \epsilon) \ln v), \tag{13}$$

$$\dot{u} = v - 1, \tag{14}$$






$$\dot{v} = -\gamma^2 \left[ u + \frac{1}{\xi}(\Theta + \ln v) \right], \tag{15}$$

where $\gamma = \sqrt{\frac{k}{M} \frac{L}{v_0}}$ is the non-dimensional frequency, and $\xi = \frac{kL}{A}$ is the non-dimensional spring constant (see also Gu et al. (1984)). Shear stress $\tau$ is derived by multiplying slip with $-\xi$ (e.g., Rice and Tse, 1986).

Finally, internal parameter $\epsilon = \frac{\sigma_n'(b-a)}{\sigma_n' a}$ is a key parameter that measures the sensitivity of the velocity relaxation and includes

$A = a\sigma_n'$ and $B = b\sigma_n'$, where $\sigma_n'$ is the effective normal stress as demonstrated in Erickson et al. (2008).

When compared to a slip weakening friction formulation, the parameter $b - a$ takes the role of a stress drop, while $a$ corresponds to the strength excess (Fig. 1b).

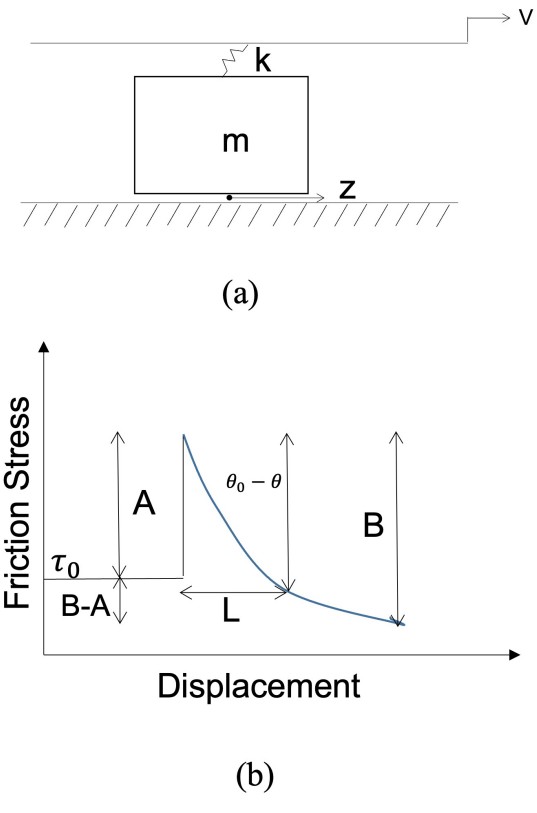

(a)

(b)

**Figure 1.** (a) zero-dimensional (0D) spring block representation used in this study. $k$ is the spring constant, $v$ the slip velocity, $z \, (= u)$ the slip, (b) Schematic diagram taken from Erickson et al. (2008) illustrating the stress response to a step change in the imposed velocity $v$ of a single spring-block slider model.



### 2.2.2 Friction parameter $\epsilon$

To study stick-slip behavior, we limit our 0D model to a velocity-weakening regime for which $a - b < 0$ and the system

generates a stable limit-cycle solution with periodic oscillations (Fig. 2). In nature, earthquakes are far from being periodic, but since we have simplified our model to a 0D system and we are not considering the spatio-temporal correlations of the faults, the system generates periodic cycles. The characteristics of the earthquake cycle are strongly influenced by friction parameters $a$ and $b$. In this study, we address these together and introduce a bias in the prior friction parameter $\epsilon = \frac{b-a}{a}$. The parameter values in this study are based on laboratory experiments by Niemeijer and Vissers (2014) done on phyllosilicate-rich fault

rocks, where we select the results representative for depths of 6 km. At this depth, 5 repeat experiments provide a range of plus or minus one standard deviation for $b - a$ from 0.00161 to 0.0077 (Figure 7a in Niemeijer and Vissers (2014)), i.e. equivalent to $\epsilon$ 0.161 - 0.77 for $a$=0.01. In our synthetic perfect model tests, we define $\epsilon = 0.7$ as the true sensitivity of velocity relaxation. Inspired by the measured $b - a$ standard deviations, we selected three cases of parameter bias, i.e., $\epsilon = 0.72$ for a small bias of +3%, $\epsilon = 0.60$ for intermediate bias of -14%, and $\epsilon = 0.40$ for a large bias of -43%. Fig. 2 presents the state trajectories for

these three cases. The large impact of the value of $\epsilon$ on the system dynamics is evident from the large difference between the trajectory of the simulation with the large parameter bias and that of the true trajectory. All parameters used in the simulations of the 'truth' for the fault slip model are summarized in Table 1.

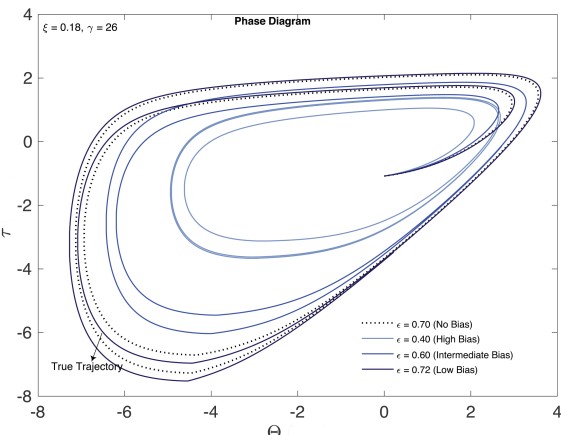

**Figure 2.** Phase diagram for a Burridge-Knopoff (BK) model with rate-and-state friction formulation when (1) $\epsilon = 0.72$ (a small bias case), (2) $\epsilon = 0.60$ (an intermediate bias case), (3) ) $\epsilon = 0.40$ (a large bias case), and (4) the model parameter is equal to the true parameter i.e. a case of no bias.





**Table 1.** Parameters of the reference model

| Parameters | Truth model parameters |
| --- | --- |
| Spring constant (k) | 10 MPa/m |
| Period of the analog freely slipping system ($2\pi\sqrt{m/k}$) | 5 s |
| Reference sliding velocity ($v_o$) | 30 mm/yr |
| Effective normal stress ($\sigma_n{'}$) | 120 MPa |
| Direct effect parameter (a) | 0.01 |
| Evolution effect parameter (b) | 0.007 |
| Non dimensional friction parameter ($\epsilon$) | 0.7 |
| Non dimensional spring constant ($\xi$) | 0.26 |
| Non dimensional frequency ($\gamma$) | 0.18 |
| Characteristic scale length (L) | 80 mm |
| Friction coefficient ($\mu_*$) | 0.60 |

# 3 Numerical experiments

## 3.1 State and state-parameter estimation for the seismic-cycle model

In state estimation in the seismic-cycle estimation, the state $\mathbf{x}$ to be estimated is given by $\mathbf{x} = [\Theta, \tau, v]^{\mathrm{T}}$. In the state- and parameter estimation, we augment this state vector with:

$$\boldsymbol{\xi} = [\epsilon],$$

where $\epsilon$ is the poorly known friction parameter. The state vector is then given by $\mathbf{z} = [\mathbf{x}, \boldsymbol{\xi}]^{\mathrm{T}}$.

By updating this state vector, we update both the state and the parameter at each new analysis time.

## 3.2 Experiment design

In the following, we consider a 'truth run' with $\epsilon_t$ and three different priors with $\epsilon_m$: one with a small bias, one with a medium bias, and one with a large bias. To test if data assimilation is indeed capable of estimating the state trajectories, we assimilate synthetic observations of both shear stress and slip rate every 4 time steps. The reason that we assimilate so frequently, is to obtain sufficient information of the behavior of the fault in the short co-seismic phase, i.e., the phase of the earthquake itself. The performance of the data assimilation is evaluated by comparing state estimates to the true trajectory.

## 3.3 Twin experiments

Since the true state of the fault system is unknown, we validate our data assimilation algorithms using twin experiments. Using these synthetic experiments, we can assess whether data assimilation methods are able to correctly evaluate the posterior





distribution of the state. For the synthetic truth we select a forward model run that follows Eq. 13, 14 and 15. The initial

conditions of the three state variables are $\Theta_0$, $d_0$, $v_0$ = (0,6,1).

For state- and parameter estimation, a different approach has been adopted to generate the prior. As we observe that the state variables are highly sensitive to the parameter value, we make sure that prior state variables are fully consistent with the prior parameter values. For this, we follow the following procedure: (1) We sample $N$ prior parameters $\epsilon'_m$ (where N is the number of particles) from a log-normal distribution of std. deviation 0.01 and mean as $\epsilon_m$.

(2) For each $i = 1,\ldots,N$ parameter values, we run the model forward for a small time duration (around 100 time units) using $\epsilon'^{,i}_m$.

(3) We sample the initial conditions of the state variables ($\Theta$, $d$, $v$) from these $N$ forward runs at any random time from each trajectory $i$.

In this way, each state variable and the parameter chosen in the ensemble ($\Theta_i$, $\tau_i$, $v_i$, $\epsilon_i$) represent a different trajectory, for

a different value of $\epsilon_m$. Since each particle has a different parameter value, the prior ensemble contains different recurrence times for earthquake slip events.

Synthetic observations are produced by sampling from the synthetic truth and adding an observational error from a Gaussian distribution with standard deviation $\sigma_\beta$.

### 3.3.1 Data assimilation settings

The experiments are performed with 1000 particles using a sequential importance resampling (SIR) particle filter. The observations are provided for the two state variables $\tau$ and $v$, as these are most likely to be observable, following the approach and large uncertainties determined in Van Dinther et al. (2019). We sample observations at intervals of 4 time units i.e. $\Delta t$ = 4. and add synthetic observational noise sampled from a normal distribution of zero mean and variance $\sigma_\beta{}^2 \sim$ N(0, $\sigma_\beta{}^2$). $\sigma_\beta$ is 0.6 for fault shear stress and 1.15 for slip velocity observations. These values are consistent with estimates of Van Dinther et al.

(2019) and Mowafy and Bilbas (2016). The stochastic model error $\eta_t$ (Eq.1) for the state evolution is sampled from a normal distribution $\sim$ N(0,$\sigma_\eta{}^2$) with $\sigma_\eta$ = 0.01. We add a value of 0.01 amplitude to represent model error in the fault shear stress $\tau$ and state variable $\Theta$ and 0.5 for slip velocity $v$.

## 4 Results and analysis

### 4.1 Case A: State estimation

Figs. 3 (b)-(d) illustrate the evolution of the posterior mean $\tau$ (red) and all prior particles (grey) in the three cases of parameter bias and compare it with the reference case of no bias ([Fig.3 (a)]). We refer to the posterior mean as the analysis. When the parameter bias is small, the prior estimates of the fault stress and slip rate evolution capture the truth well as seen from Fig.3 (b). On the other hand, for the case of intermediate (Fig.3 (c)) and large bias Fig.3 (d), the prior state estimation fails to capture the evolution of the shear stress after a few seismic cycles, which complicates the data assimilation going forward. Fig. 2(b)





clearly demonstrates that the data assimilation in the large bias case ($\epsilon_m = 0.40$) is unable to account for the difference in state
evolution caused by the parameter bias.

To better understand and improve these data assimilation results, we analyze the results for the experiment with intermediate
bias in more detail (Fig. 4). By assimilation of data representative of both the interseimic and coseismic periods, we determine
whether stress can be estimated. Throughout the paper we illustrate the stress for a number of assimilation times that represent

the time epochs before and after an earthquake (Fig. 3a): $t = 16, 20, 24, 72, 76$. In the enlarged version of the figure (Fig. 5), we
can see the probabilistic estimate of fault shear stress based on assimilation of coseismic data at $t = 20$. The prior and posterior
density represent the particles before and after undergoing resampling respectively. The mean value of the posterior density
of stress corresponds better to the true value than that of the prior density. In fact, at time $t = 20$, the particles closest to the
observation (representing a pre-slip condition) have the highest likelihood and consequently obtain the highest weights. As a

consequence, the posterior density has its peak close to the truth. It should be noted that particles with shear stress values that
are farther away from the observations may still obtain high weights because of their fit to the observed slip rate. As a result,
the posterior trajectory is close to the observations, but fails to completely follow the true trajectory. Fig. 6 (a)-(j) illustrates
this further with phase diagrams, which show by means of scatter plots the distribution of particles in the $\Theta$ - $\tau$ space. The blue
and black line in the phase diagram of Fig. 2(b) represent the trajectory of the particles when the model parameter $\epsilon_m$ is 0.60

(intermediate bias) and 0.70 (no bias), respectively.The size as well as the color of the symbols indicate the particle weights.
Fig. 7a-j presents the resulting PDF distribution of the corresponding plots in Fig. 6.

The particles with a bias in parameter show a phase difference compared to the truth (Fig. 6a-e). Trajectories of particles
with lower values of $\epsilon$ have a shorter cycle and their $\Theta - \tau$ trajectories are always ahead of the true trajectories. At time $t = 16$,
i.e., in the inter-seismic phase, the particles and the truth are almost in sync. However, at the co-seismic phase ($t = 20$), minor

differences in the state for that specific moment can result in large differences in the state trajectory. In this phase, particles that
were close to each other previously are pulled apart. After the co-seismic phase, i.e., for $t = 24$, the particles of the biased prior
parameter, which have shorter cycles than the truth, reach the inter-seismic phase of the subsequent earthquake event faster
than the truth. After this, the pattern repeats itself. When there is no bias (Fig. 6(f)-(j)), the particles are always in sync with the
truth. Compensating for a bias in the prior parameter could also be achieved by changing assimilation settings. In this study,

we explore the effect of (i) increasing the model error, or/and (ii) resampling twice rather than once.

Using different assimilation settings, it is possible to inflate the ensemble.

We note that a biased parameter value makes it physically impossible to have $\Theta$ and $\tau$ in the same phase and amplitude as in
the true seismic cycle for an extended period of time. With different assimilation settings, though, it might be feasible to adjust
the state such, that we can correctly predict the arrival of the next earthquake. Fig.8a-c shows the state-estimation results for

three different data assimilation settings as well as the case of joint state-parameter estimation in Fig.8d. It can be seen from the
figure that increasing model error along with frequent resampling leads to a more effective compensation of the biased prior
parameter than state estimation without these settings (Fig.3c).



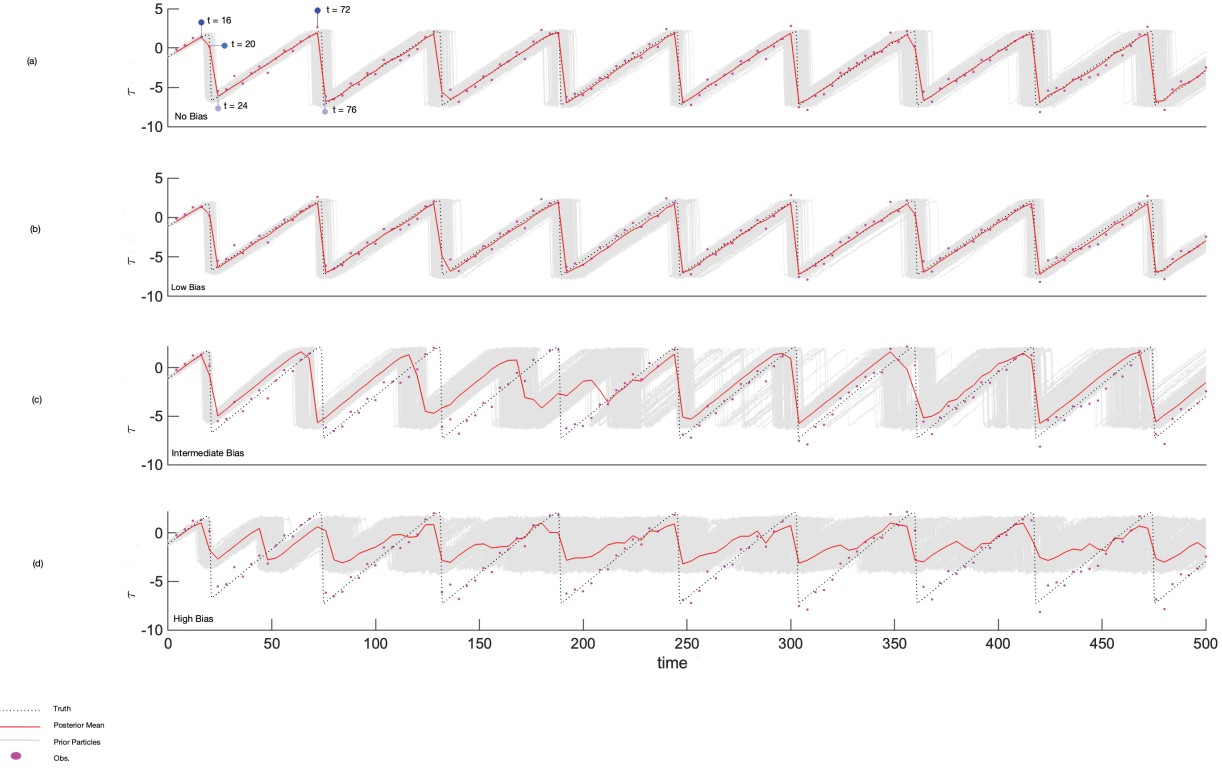

**Figure 3.** Evolution of 1000 particles along with the posterior mean, observation and truth of state variable $\tau$ for state estimation for (a) the case when the model parameter is equal to the true parameter i.e. a case of no bias ($\epsilon_m = 0.70$), (b) a small bias case ($\epsilon_m = 0.72$), (c) an intermediate bias case ($\epsilon_m = 0.60$) and (d) large bias ($\epsilon_m = 0.40$), when $\epsilon_t = 0.70$, $\gamma = 0.26$ and $\xi = 0.18$.



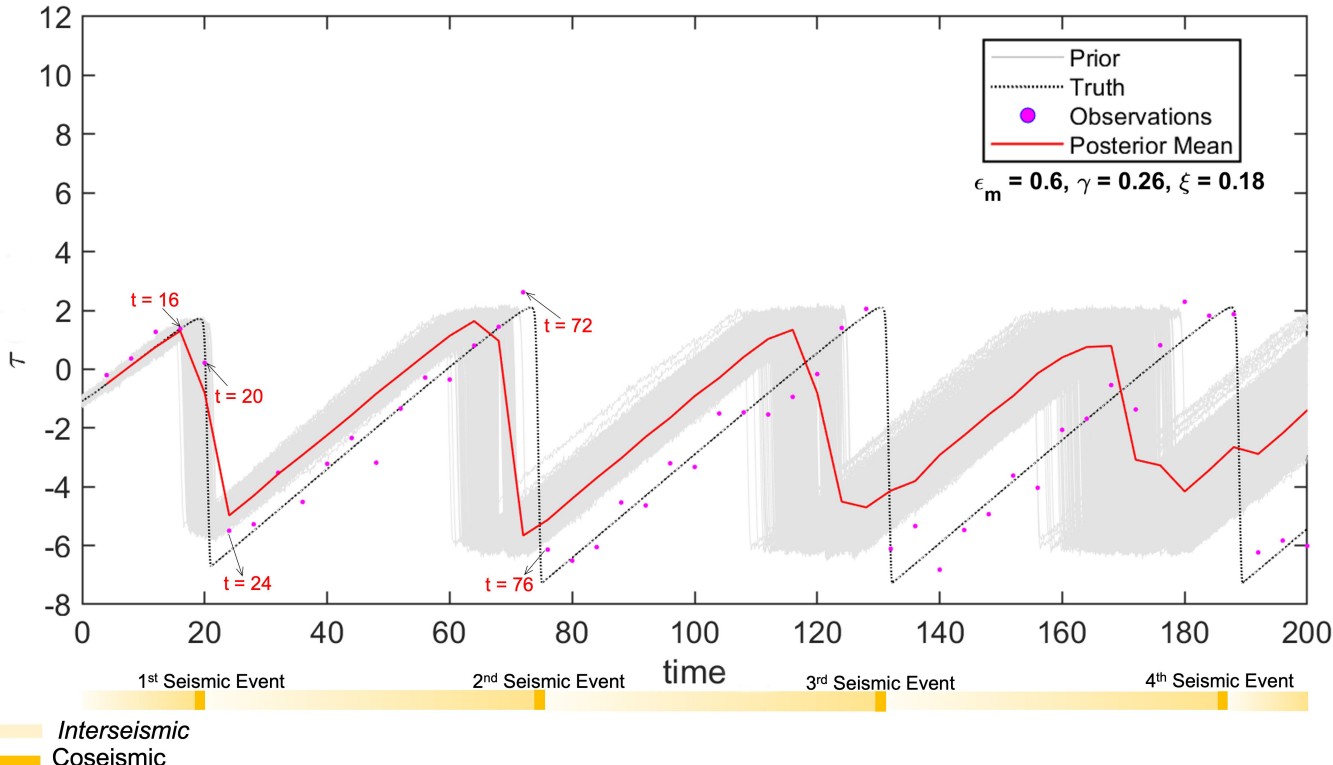

**Figure 4.** Shear stress evolution. The coseismic and inter-seismic phases in the seismic cycle have been highlighted. The time points discussed in the study are $t = 16, 20, 24, 72, 76$ in the interseismic phase and $t = 20$ in the coseismic phase.



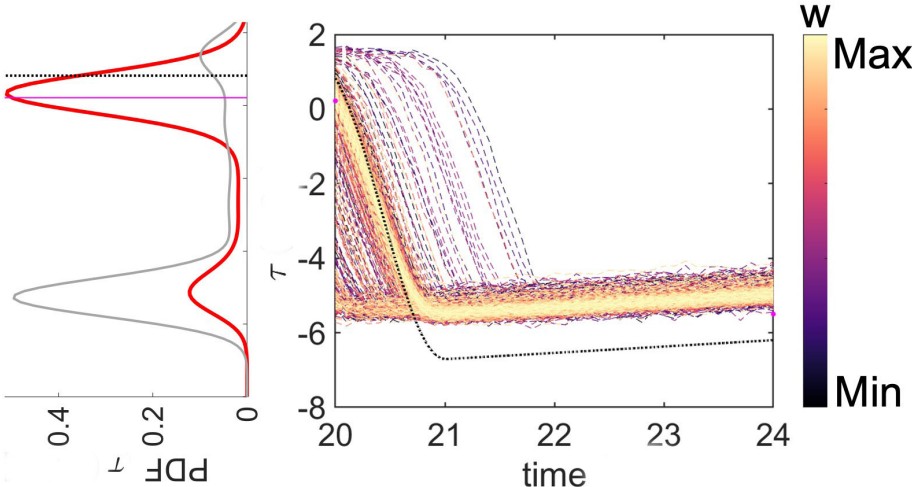

**Figure 5.** Shear stress evolution with weight distribution at time $t = 20$. The color of the lines represents the value of the weight, which is a function of the distance between the prior shear stress estimate and the observation

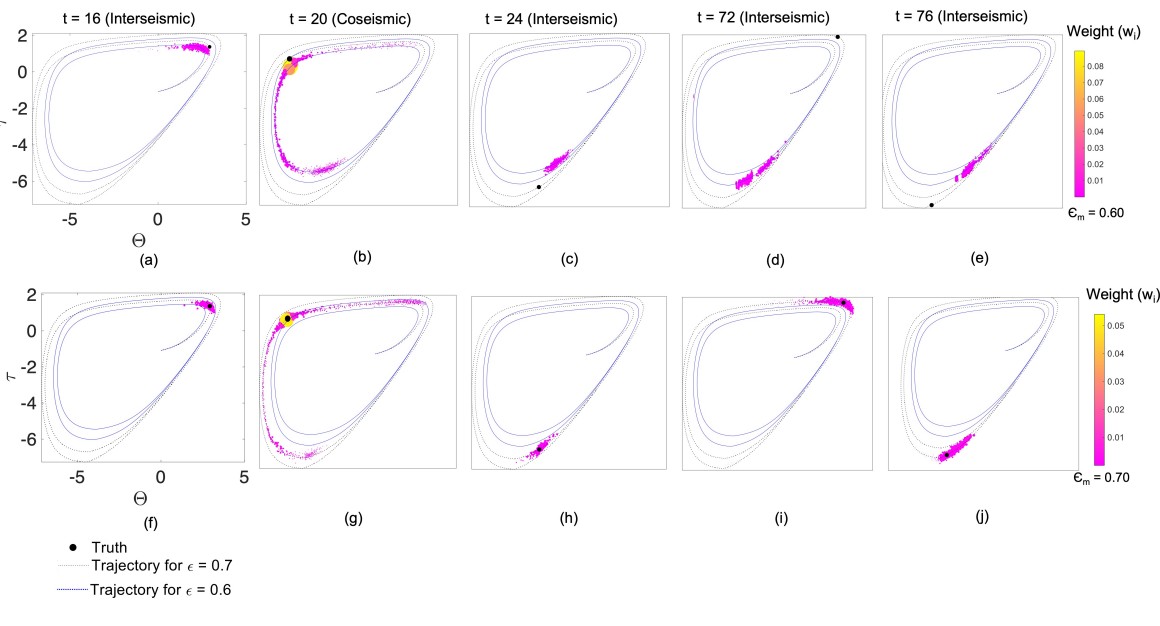

**Figure 6.** Particles in the $\Theta$-$\tau$ space for the case of state estimation with (a-e): intermediate bias ($\epsilon_m = 0.6$) and (f-j) no bias ($\epsilon_m = \epsilon_t = 0.7$). The points represent the particles in the distribution while the lines represent the trajectories in the phase diagram for the respective value of the parameter $\epsilon_m$. The blue line represents the phase diagram for $\epsilon_m = 0.6$ whereas the black line is for the case when $\epsilon_t = 0.7$. The size and color of the symbols representing the particles reflect the weight of each particle.





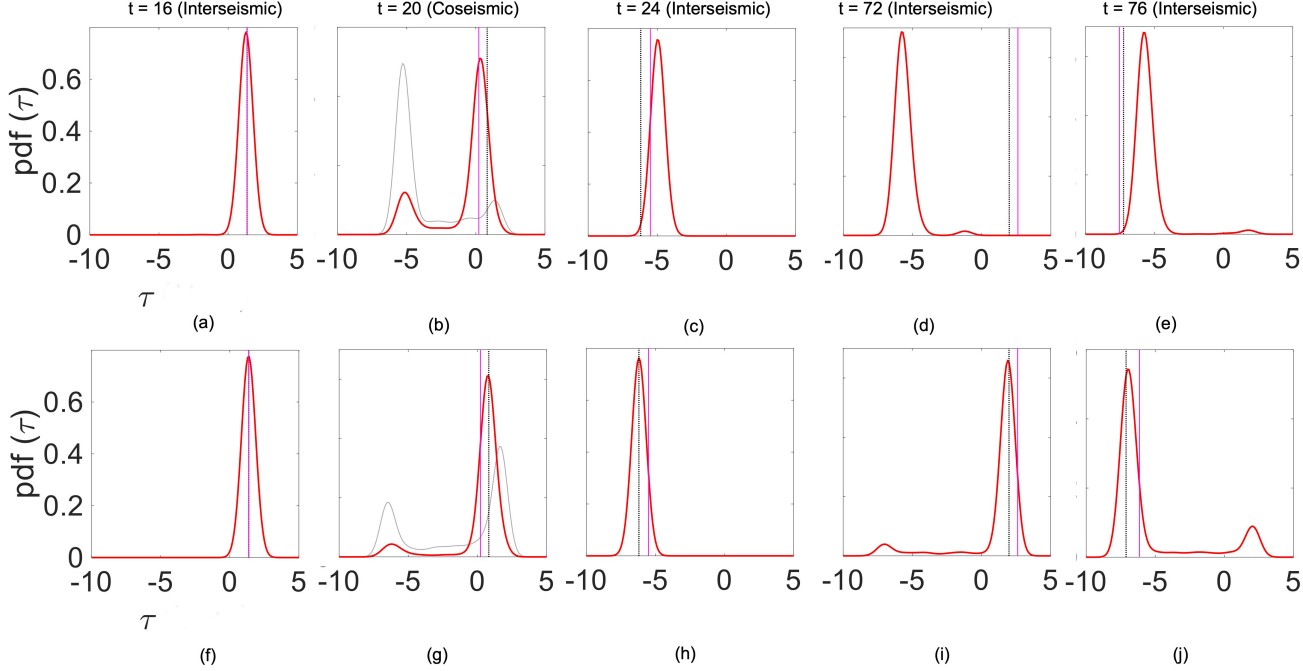

**Figure 7.** Corresponding PDFs representing the posterior mean in red, prior in grey line, observation in magenta and truth in black line for the case illustrated in Fig.6.




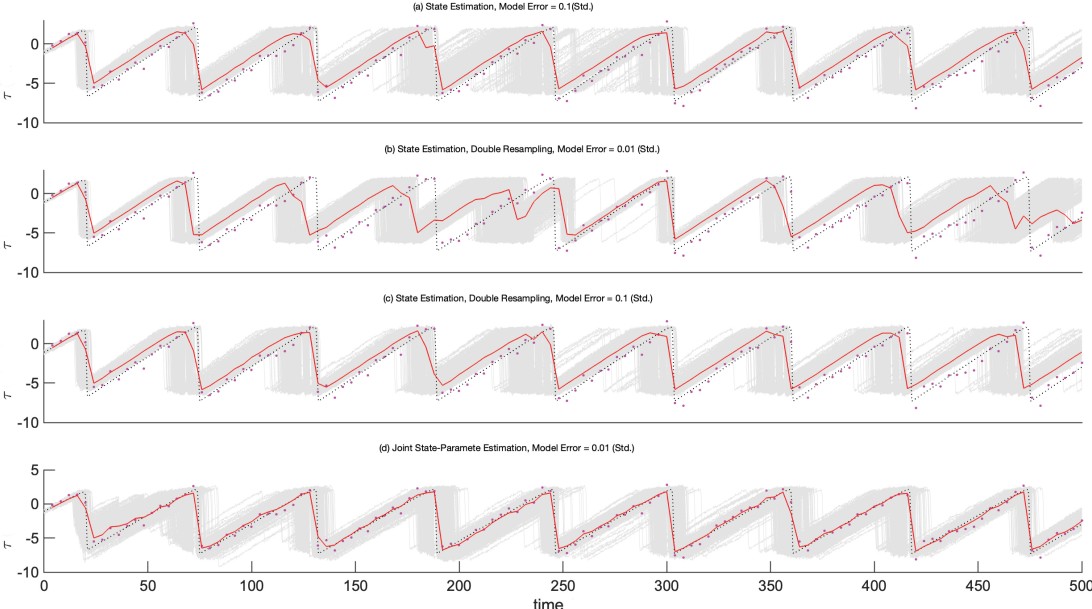

**Figure 8.** Evolution of particles with time for intermediate bias in state estimation with (a) higher model error ($\eta_t = 0.1$), (b) double resampling ($\eta_t = 0.01$), (c) combination of higher model error ($\eta_t = 0.1$) and double resampling. (d) Joint state-parameter estimation ($\eta_t = 0.01$).

## 4.2 Case B: Increased model error

The model error in the forward-model equation represents the imperfections in the model and thereby maintains ensemble
spread. Adding $\eta_t$ to the forward model equations improves the efficacy of the filter. Choosing a smaller model error reduces the spread, which restricts the solution space considerably.

We find that inclusion of model error has a noticeable positive effect on the posterior estimate (brings the mean of the posterior closer to the truth) when the parameter bias is either small or intermediate.

We observe that increasing the model error causes the prior $\tau$ distribution of the particles to have two peaks in the co-seismic
phase. This was a single peak when the model error was smaller. This shows that increasing model error allows for a larger variety of states in the prior. The phase diagrams for the intermediate bias case when we increase the model error ($\eta_t$ in Eq.1) is presented in the supplement figures (Figure S1).

## 4.3 Case C: Double resampling

Resampling improves the effectiveness of the particle filter by introducing more particles with a state close to the observed
state 2.1.1. We observe that double resampling retains particles that are in the same interseismic phase as that of the truth as in $t = 24$). From this, we conclude that double resampling can be useful in retaining more particles "in sync" with the truth in





the distribution, but additional spread in the ensemble is required to shift the posterior distribution more towards the truth. The effect of double resampling on the posterior distribution for an intermediate bias is provided in the supplement figures (Figure S2).

### 4.4 Case D: Increased model error and double resampling

We study the effect of increasing the model error with consequent double resampling on the posterior estimates and use the term 'combined adjustment' for this data assimilation setting. The posterior distribution of stress for the combined adjustment captures the truth distinctly better compared to the case with model error only. There is no significant difference in the forecasting ability of the prior particle distribution in the first seismic cycle for the combined adjustment. Increasing model errors and double resampling both increase variability within the particles. However, at time $t = 76$, after the co-seismic phase, particles of this combined adjustment experiment are no longer fully in sync with the truth. The effect of combined adjustment on the posterior distribution for an intermediate bias is provided in the supplement figures (Figure S3).

### 4.5 Case E: Joint state-parameter estimation

Fig.9 presents the results for state-parameter estimation in phase diagrams. Fig.10 presents the PDF distribution of the phase diagrams. Not surprisingly, joint state-parameter estimation (Fig.9 a-e improves the posterior distribution compared to the case of state estimation (Fig.9 f-j.

The parameter estimate gradually reaches nearly its true value after a certain time period. In the state-parameter estimation case, the parameter estimate gradually changes from its prior, biased, value towards the correct estimate of $\epsilon = 0.7$ and remains at that value after that. Hence, in this case, the state variables are improved in the next seismic cycle which is not the case in state estimation. From this we can conclude that joint state-parameter estimation is in this case the most effective approach for seismic cycle estimation in the presence of a parameter bias. It is important to note that the state-parameter estimate does not only lead to a reasonably good posterior estimate of the state at these selected time steps, but at all assimilation time steps in the period considered, i.e., from $t = 4$ to $t = 500$ (Fig.4).




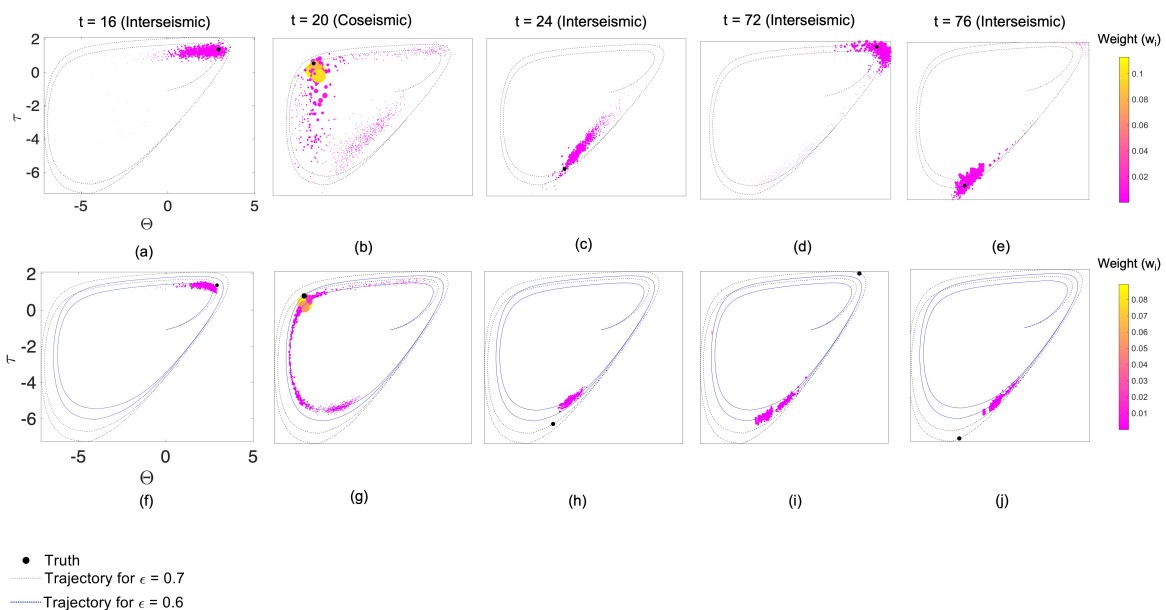

**Figure 9.** $\Theta - \tau$ phase diagrams for the case with (a-e) state-parameter estimation and (f-j) state estimation for intermediate bias. Use of colors and lines is the same as in Fig. 6.

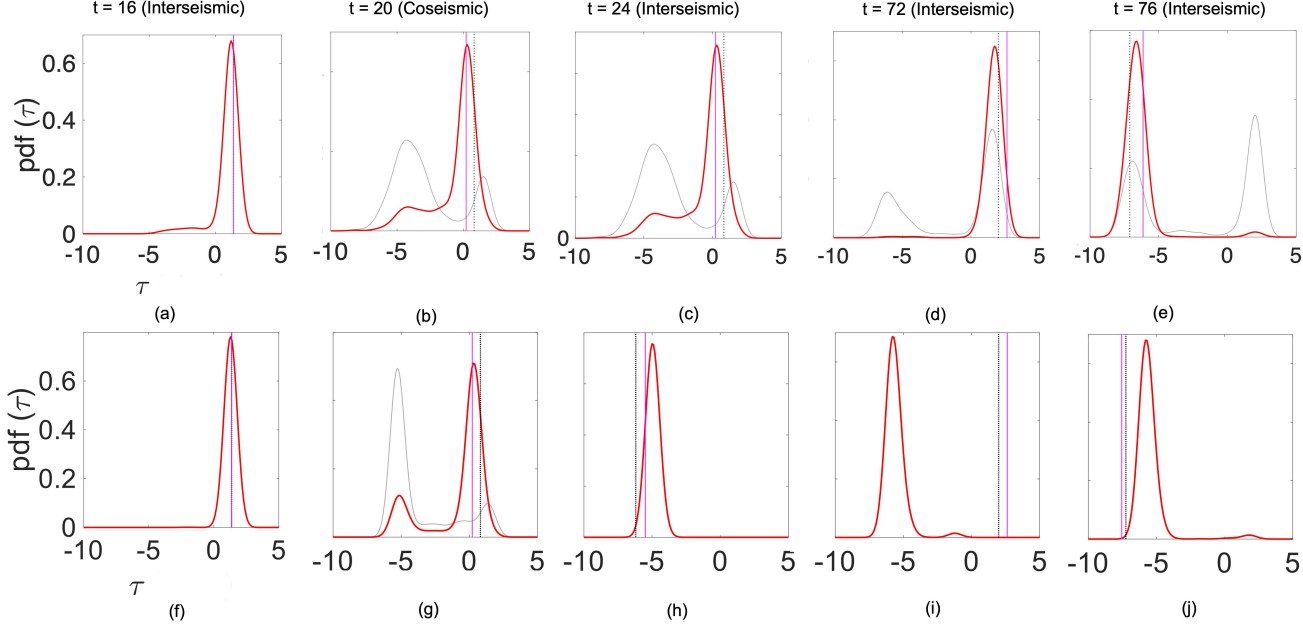

**Figure 10.** Corresponding PDFs for the case with (a-e) joint state-parameter estimation and (f-j) state estimation with intermediate bias in Fig. 9.



To evaluate the overall accuracy of the state- and the joint state parameter estimation for the different biased-parameter cases,
we perform a regression analysis. The results in Fig.11 show how close the posterior estimates of the different experiments
are to the truth using the R-squared ($R^2$) value. We find that when the parameter bias is small, state estimation is efficient
with an $R^2$ value of 0.99. For the case of the intermediate bias, the setting of increasing the model error to 0.1 and consequent
resampling with a smaller model error of std 0.01, results in an $R^2$ value of 0.61. However, for all of the cases, the joint state-
parameter estimation leads to significantly higher $R^2$ values and hence improved state estimates compared to the cases with
state estimation only. This confirms our conclusion from section 4.5.

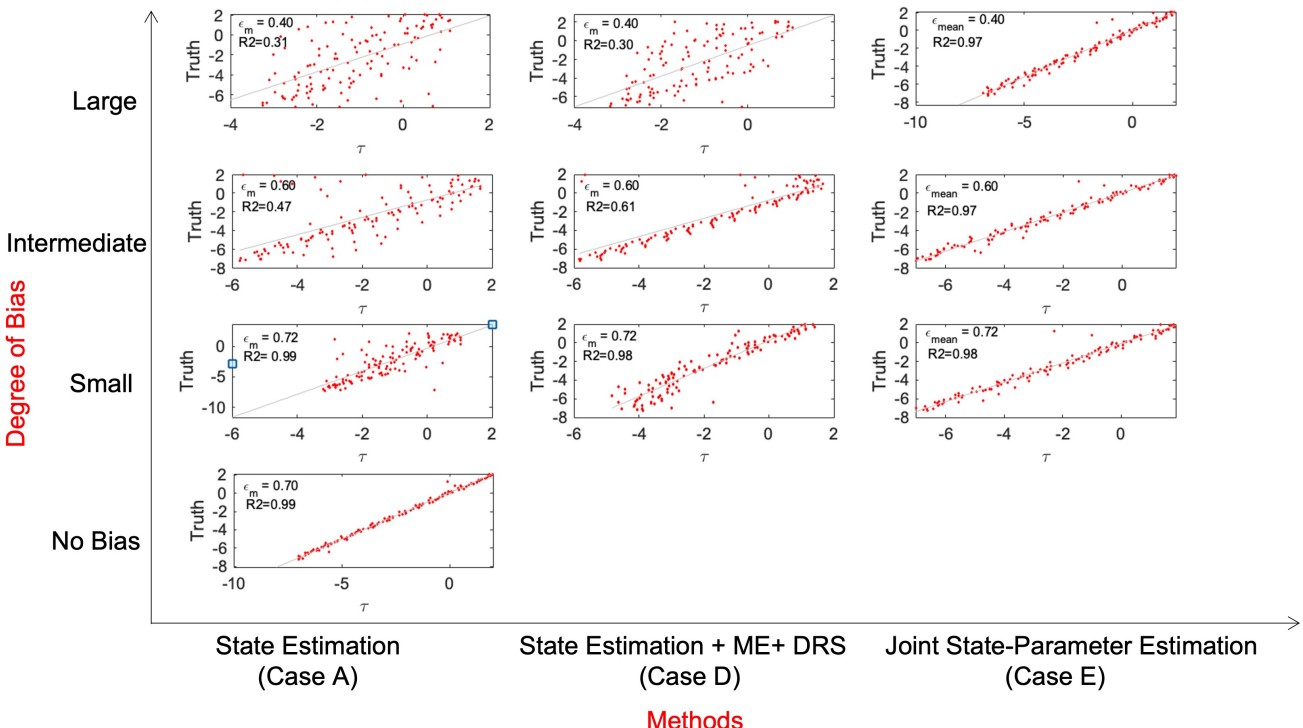

**Figure 11.** Comparison of regression between the estimate of $\tau$ and the true $\tau$ and the corresponding $R^2$ values for the three bias cases
(rows) and for different assimilation approaches (columns). The red dots indicate the estimated values or the posterior and true $\tau$ values at
the observation times. A diagonal line is added for reference.

## 5  Discussion

The results of this study illustrate the implications of parameter bias in data-assimilation applications for seismic-cycle mod-
eling. They show it is feasible to apply a particle filter to a fault slip model. Moreover, they reveal that uncertainties in friction
parameters can be accounted either via directly estimating these parameters or via a larger model error. Accounting for these
improved parameter estimates, also fault state estimates can be improved. This demonstrates that a possible trade-off between



estimating shear stress and friction can be effectively accounted for using data assimilation methods. In other words, data assimilation is able to separate error contributions due to uncertainties in friction (i.e., shear over normal stress) from uncertainties in estimates of a fault's shear stresses. This suggests that potentially large uncertainties in friction do not hamper the further development in directions using data assimilation methods or for physics-based seismic hazard assessment.

The simplest and most effective approach to deal with parameter bias, depends on the degree of parameter bias observed. For laboratory experiments with increased accuracy, one can potentially still use and tune state estimation, e.g., through increasing model error. However, for observed uncertainties in laboratory experiments in Niemeijer and Vissers (2014), and particularly when also accounting uncertainties in in-situ conditions (e.g., effective pressure and thermal conditions and amount of displacement) and laboratory setup (e.g., sample type with or without gouge) as well, state-parameter estimation is clearly the

best and the required approach. It is remarkable to observe that by accounting for state-parameter estimation, any degree of bias can be dealt with equally effectively (Figure 11). This suggests that even upon extrapolation to significantly larger biases, state-parameter estimation can be an effective solution for earthquake cycle problems.

    It is important to remember that these findings are based on the performance of state update and state-parameter update algorithms for a simplified nonlinear physics-based model in synthetic experiments. Some limitations that are still present in

this pioneering study, which eventually need to be addressed, include:

- In this study, we assimilate both the fault shear stress and the slip rate observations on the plate interface. However, we have also conducted experiments with only assimilating the fault shear stress observations. It is observed in this case that the state estimate yields better fitting to the truth compared to the case when we assimilate both types of observations or only the slip rate. Assimilating slip-rate observations leads to lower weight values owing to their relatively high

observation error. This affects the posterior estimate. However, a thorough study still needs to be conducted to identify which observations provide the most meaningful information when assimilated in earthquake-cycle models.

- In this study, we focus only on the uncertainty with respect to parameter $\epsilon$. It can be seen from Eq.13, 14 and 15 that also the highly uncertain characteristic slip length L impacts fault states and cycle characteristics.

- Another point of attention is the selection of the number of particles required for a correct sampling of the prior. Re-

alization of 1000 particles may be computationally expensive for models that are more complex than the model used here. Improved sampling techniques, for example using a proposal density function, or using a particle flow (e.g. Hu and Van Leeuwen, 2021) could help to reduce computational costs without loosing the advantages of a nonlinear filtering methods.

- In the present work, we investigated a simplified version of a Burridge-Knopoff spring-block slide model in a simplified

0D form. Eventually, to accurately simulate the behavior of real earthquake faults, 1D, 2D and 3D simulation models that allow for more spatial heterogeneities will need to be included Li et al. (2021).

Nonetheless, simplified models provide useful insights to solve more complex problems. This is an important stepping stone to the development of data-assimilation applications for the simulation of more realistic earthquake cycles. Future developments



for these purposes addressing the above limitations, could utilize other methods explicitly accounting for parameter-bias in the
data-assimilation scheme to obtain a better accuracy (e.g., Dee and Da Silva, 1998; Sørensen and Madsen, 2004; Chepurin et al., 2005; Auligné et al., 2007; Li et al., 2009; Du et al., 2020).

## 6  Conclusions

In this study, we demonstrated the effect of a parameter bias in an earthquake cycle model on the estimated states with data assimilation. Synthetic, noisy observations of the shear stress and slip velocity on the fault plate interface were assimilated with state and state-parameter estimation methods using a particle filter. In our forward model, the shear-stress estimates strongly depend on the friction parameter $\epsilon$. Therefore an inaccurate representation of this parameter would impact the stress estimates and forecasted cycles obtained using this model. To quantify this impact, we considered three different magnitudes of biases with respect to the true parameter $\epsilon_t = 0.7$, i.e., (i) small bias (model parameter $\epsilon_m = 0.72$), (ii) intermediate bias (model parameter $\epsilon_m = 0.60$) and (iii) large bias (model parameter $\epsilon_m = 0.40$).

We find that for a small bias in friction parameter $\epsilon$, state estimation is most effective ($R^2 = 0.99$ for the regression between estimated and true shear stress). In the case of an intermediate bias, state estimation alone is less effective ($R^2 = 0.47$). However, by increasing the prior model error and adding a second resampling step in the data assimilation approach, the results can be improved ($R^2 = 0.61$). Nonetheless, state-parameter estimation in this case is best ($R^2 = 0.97$). In the case of a large parameter bias, state-parameter estimation ($R^2 = 0.97$) is significantly more effective manner to reconstruct the true state than state estimation ($R^2 = 0.31$) or improved state estimation with combined adjustments ($R^2 = 0.30$). This suggests that state-parameter estimation using data assimilation could be an effective method to improve forecasts of frequently recurring fault slip events.

*Data availability.* The observations in the study were generated by simulation. Extra explanatory figures have been uploaded to the Supplement.

*Acknowledgements.* The contributions of FV and AB have been funded by the Delft Technology Fellowship of Delft University of Technology. This work has been funded by DeepNL InFocus project funded by NWO (DEEP.NL.2018.037). The work benefited from discussions with André Niemeijer of Utrecht University and Hamed Diab-Montero of Delft University of Technology.





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
