# Peer review of "On Parameter Bias in Earthquake Sequence Models using Data Assimilation"

_EGUsphere, 2022_

## Author Comment (AC1)

Response to the editorial comments for the Manuscript entitled "**On Parameter Bias in Earthquake Sequence Models using Data Assimilation**."

The authors are thankful to the editor for his insightful guidance in making us improve the quality of the manuscript and for considering us for possible publication in the esteemed journal of 'Nonlinear Processes in Geophysics'**.** The authors are thankful to the reviewers for their helpful comments. Following the comments of the reviewers, the necessary changes are made in the revised manuscript.

**Comments from Reviewer 1**

Comment 1:
Synthetic observations are produced by sampling from the synthetic truth and adding an observational error from a Gaussian distribution with standard deviation. However, the real observations could be affected by instrumental noise, missing data, spikes, etc, and a short time step of four time units may no longer be applicable. I understand that the authors lack real observations. But they should at least discuss this limitation in Section 5.

Response:
The authors thank the reviewer for the comment. Indeed, the assumption of data availability (once every four-time units) and the assumptions on the standard deviation and distribution of the observational error may not be valid when assimilating real data. We have mentioned this as a limitation in our discussion (page 21, line 348-356) as:

*An additional point worth mentioning is the use of synthetic observations for fault displacement and velocities for data assimilation in this study. In realistic applications, the assumptions that we have considered with respect to data assimilation frequency and the standard deviation and distribution of the observational errors may not be valid. However, if we know the distribution of the measurement errors, we can use that information in choosing the relevant likelihood function that can greatly affect our fault estimates. Fault shear stress observations are usually not available, and if they are, they are subject to large errors. In contrast, fault velocities can be observed fairly accurately using GPS, as discussed by Van Dinther et al. (2019), who demonstrate that stress measurements are useful despite of their large errors. Following van Dinther et al (2019), we emphasize the need for additional sensitivity studies to understand the implications of data gaps, outliers, and instrumental noise before our proposed methods can be used on real data.*

Comment No. 2:

In introduction, please review some previous studies where either the frictional parameters have been estimated as part of the data assimilation or assumed to be perfectly known.

Response:
The authors have included some references which estimates frictional parameters using data assimilation and are mentioned in the manuscript as (page 2, line 35-37):

*'On the other hand, Van Dinther et al. (2019) and Diab-Montero et al. (2022) assumed the frictional parameters to be known and used an ensemble Kalman filter to estimate the fault states.'*

Comment No. 3:

Section 2.1: Please specify how to quantitively determine the observation noise error. Please review some data assimilation frameworks and explain the reason why this framework is selected.

Response:
Observational errors can be determined by comparing observations of velocity or displacement with independent observations of these variables. This text is added to the manuscript. In the introduction, we motivate the choice for using ensemble-based data assimilation methods especially particle filters (page 2, line 38-44). For a further review of data assimilation frameworks, we refer the reader to the recent book by Evensen et al (2022). This reference has been added to the respective text in the introduction as (line 40-41, page 2):

*For further discussion of available data-assimilation methods, we refer the reader to Evensen et al. (2022).*

Comment No. 4:

Equation 4: What does j mean? Is it a typo?

Response:
The authors appreciate the reviewer's comments. It is a typo and it should be i which is the number of realization. It has been corrected in the manuscript.

Comment No. 5:
Line 89: In the presence of filter degeneracy, how to guarantee that one or few particles with high weight are sufficiently representative as the input?

Response:
In degeneracy, the weight of one particle is close to one while the weight of all the other particles is close to zero. In this case, a single particle represents the filtered distribution, which results in an extremely poor approximation. Hence it is important to avoid filter degeneracy by (i) adding jitter in the prior distribution and (ii) using resampling step in particle filter. In the present work, we have included both to avoid filter degeneracy.

Comment No. 6:
Line 96: The sequential importance resampling process duplicates particles with high weight. Please explain its physical meaning in data assimilation.

Response:
The authors appreciate the reviewers' comments. In this implementation of the particle filter, the sequential importance resampling attributes higher weight to particles that are closer to the observations. This is done by multiplying the prior by the likelihood, which can be considered a weight function (in this study, a Lorentz function). Then, in the resampling step, the

importance resampling process removes those particles which have low weight in the distribution and thereby retains only those particles which have a higher weight. These are then duplicated according to their weight, in such a way that the number of particles remains constant. This ensures an approximation of the prior distribution that is less sensitive to particle degeneracy. We have mentioned this in text from line 102-105 in page 4.

**Comment No. 7:**
Section 2.2: The model of forwarding simulation is important to data assimilation. In this study, a zero-dimensional (0D) model is considered. However, 1D, 2/3D models are also available. Please specify the reason why 0D model is selected. More details of its pros and cons are expected.

Response:
The authors appreciate the comment of the reviewer. We have explained the reason of choosing a simplified model for this study in detail from line 357-366 (page 21) in the discussion of the manuscript as:

*It is also very important to highlight the reason behind selecting a zero dimensional (0D) model for this study. Simplified fault slip models are computational efficient tools that help us to understand the physics behind the earthquake dynamics. A study by Li et al. (2021) compares the simulation of earthquakes in 0D, 1D, 2D, and 3D models and finds that lower-dimension models (0D and 1D) qualitatively represent the same dynamics as 2D and 3D models. Although 0D models cannot simulate the full complexity of the earthquake physics, they have the advantage that they are computationally inexpensive and provide the user with a tractable conceptual description of earthquake physics and the importance of the friction parameter. In our case, we were interested in investigating the effect of frictional parameter bias on the estimated fault states in earthquake cycle models. Hence in the present work, we investigated a simplified version of a Burridge-Knopoff spring-block slide model in a simplified 0D form. Eventually, to accurately simulate the behavior of real earthquake faults, 1D, 2D and 3D simulation models will be required (e.g., Li et al., 2021).*

**Comment No. 8:**
Line 149: What if in the region a-b>0?

Response:

The parameter combination $(a - b) < 0$ corresponds to steady state slip rate-weakening properties causing an unstable rapid slip (frictionally unstable), while $(a - b) > 0$ corresponds to the steady state velocity- strengthening behaviour, causing a stable slip (frictionally stable). Since we are primarily interested in frictionally unstable earthquake cycles, we have focused on parameter combination for $(a - b) < 0$. According to Ruina (1983), if a velocity-strengthening system experiences a slip instability, the motion will be rapidly dampened down to a state of stability. A velocity-weakening system, on the other hand, will no matter how carefully driven, always exhibit growing oscillations and reach a state of regular stick slip (Scholz (2019)). The velocity-strengthening behaviour is thus intrinsically stable. For this reason, we have not investigated the case of $(a-b) > 0$ in this study.

Comment No. 9:
Section 3.2: The assimilation step may have an important effect on the results. In this study a very short time step is adopted. Please provide more discussions on its effect. If a longer time step is used, can a small parameter bias still be compensated?

Response:

Having large assimilation steps can also have a detrimental effect on the data assimilation process as it can miss characteristic variations of the earthquake cycle. A parameter bias can have a substantial effect on the evolution of the state variables, which may be difficult to correct if the assimilation step is large. Hence a short time step is to be chosen that allows the assimilation to capture the important characteristics of the earthquake cycle.

Comment 10:
Discussion: I appreciate the authors' efforts in stating the limitations of this study, but here I expect more discussion on their results and comparison with previous studies (without data assimilation).

Response:

We extended the discussion with a section that discusses our results in relation to previous studies without data assimilation (line 330-339, page 21) as:

*Typically, earthquake forecasting is approached in a probabilistic manner (e.g., Marzocchi et al., 2017). Kinematic inversions of earthquake global positioning system (GPS) data have been used to estimate frictional properties in afterslip areas (Miyazaki et al., 2004; Hsu et al., 2006), but not for estimation of the earthquake dynamics themselves. As outlined by Van Dinther et al (2019), data assimilation for earthquake sequences has the advantage that it can take into account measurement and model errors, non-Gaussian probabilities and sequential updating as data becomes available. The results of this study demonstrate how, for a highly simplified representation of earthquake cycles, non-linear data assimilation provides a means to account for both measurement errors and parameter biases. It also highlights how observations can be included as they become available. While particle filters are not computationally efficient, they can propagate the full error distribution which makes them attractive for estimation and forecasting of highly nonlinear processes like earthquake generation.*

**References:**

Evensen, G., Vossepoel, F. C., & van Leeuwen, P. J. (2022). Data Assimilation Fundamentals: A Unified Formulation of the State and Parameter Estimation Problem.

Hsu, Y. J., Simons, M., Avouac, J. P., Galetzka, J., Sieh, K., Chlieh, M., ... & Bock, Y. (2006). Frictional afterslip following the 2005 Nias-Simeulue earthquake, Sumatra. *Science*, *312*(5782), 1921-1926.

Li, M., Pranger, C., & van Dinther, Y. (2022). Characteristics of Earthquake Cycles: A Cross-Dimensional Comparison of 0D to 3D Numerical Models. *Journal of Geophysical Research: Solid Earth*, *127*(8), e2021JB023726.

Miyazaki, S. I., Segall, P., Fukuda, J., & Kato, T. (2004). Space time distribution of afterslip following the 2003 Tokachi-oki earthquake: Implications for variations in fault zone frictional properties. *Geophysical Research Letters*, *31*(6).

Ruina, A. (1983). Slip instability and state variable friction laws. *Journal of Geophysical Research: Solid Earth*, *88*(B12), 10359-10370.

Scholz, C. H. (2019). *The mechanics of earthquakes and faulting*. Cambridge university press.

van Dinther, Y., Künsch, H. R., & Fichtner, A. (2019). Ensemble data assimilation for earthquake sequences: probabilistic estimation and forecasting of fault stresses. *Geophysical Journal International*, *217*(3), 1453-1478.

---

## Author Comment (AC2)

Response to the editorial comments for the Manuscript entitled "**On Parameter Bias in Earthquake Sequence Models using Data Assimilation**."

The authors are thankful to the editor for his insightful guidance in making us improve the quality of the manuscript and for considering us for possible publication in the esteemed journal of 'Nonlinear Processes in Geophysics'**.** The authors are thankful to the reviewers for their helpful comments. Following the comments of the reviewers, the necessary changes are made in the revised manuscript.

**Comments from Reviewer 2**

**Comment 1:**
The study uses particle filter as the data assimilation method to solve the problem. But it would be good to inform the reader which other data assimilation methods were previously used in studies related to earthquake modelling and if those were successful. If not, which were the main issues and why a particle filter would suit better in this problem compared to those. This would mainly situate the reader on the importance of your choice on the method for this study.

Response:
Several different data assimilation methods have been used previously for estimating states for earthquake models. It has been reported in the manuscript (line 25-27, page 2) as:

*'Few studies have introduced data assimilation for the purpose of earthquake forecasting (e.g., Van Dinther et al., 2019; Werner et al., 2011; Hirahara and Nishikiori, 2019; Hori et al., 2014; Llenos and McGuire, 2011)'.*

However, the data assimilation methods used in these references have not been explicitly mentioned in the manuscript. This has been modified as follows from line 32-43 in Page 2 as:

*Several studies have considered uncertainties in parameters using data assimilation in earthquake-cycle models (e.g. Kano et al., 2010, 2013; Fakuda et al., 2009; Werner et al., 2011). Kano et al. (2013) and Kano et al. (2010) used an adjoint-based data assimilation method to estimate frictional parameters of afterslip. Fakuda et al. (2009) used a Markov chain Monte Carlo (MCMC) based method for estimating fault friction parameters. On the other hand, Van Dinther et al. (2019) and Diab-Montero et al. (2022) assumed the frictional parameters to be known and used an ensemble Kalman filter to estimate the fault states. In our present study, we have used particle filters which is also an ensemble based data assimilation method and is highly efficient for non-linear systems with a non-Gaussian prior distribution. Ensemble Kalman filters are not very efficient in encountering non Gaussian characteristics. This is the reason for choosing a particle filter for this study.*

**Comment 2:**
Equations 4 and 5: What does j mean?

Response:
The authors are thankful to the reviewer for the comment. J is a typo, and it should be i which is the number of particles or realization. It has been corrected in the manuscript.

Comment 3:
The authors use a Lorentz function instead of a Gaussian to prevent filter degeneracy. In addition, a SIR step is used to further avoid this issue. Were these enough to avoid filter degeneracy or the system still presents the problem?

Response:
The authors are thankful to the reviewer for the comment. Yes, the system did not face any filter degeneracy issues. We introduce a resampling step known as sequential importance resampling (SIR). This resampling discards particles with very low weights, while duplicating particles with high weights. Additionally, we introduced jitter in the prior distribution to avoid degeneracy.

Comment 4:
The authors mention that real earthquakes are far from being periodic but, as they have considered a 0D model, their system generates periodic cycles. I wonder how far from a real state this 0D model is and why this study has not used a 1D model, in which at least a minimum spatial dimension would be considered. It will be interesting to address in the manuscript why the 0D model was chosen in this case.

Response:
The authors are thankful to the reviewer for the comment. We have addressed this comment by adding a paragraph on the use of 0D, 1D and higher models for studying earthquake cycles. It has been mentioned in the discussion as (line 357-366 in page 21 in text):

*It is also very important to highlight the reason behind selecting a zero dimensional (0D) model for this study. Simplified fault slip models are computational efficient tools that help us to understand the physics behind the earthquake dynamics. A study by Li et al. (2021) compares the simulation of earthquakes in 0D, 1D, 2D, and 3D models and finds that lower-dimension models (0D and 1D) qualitatively represent the same dynamics as 2D and 3D models. Although 0D models cannot simulate the full complexity of the earthquake physics, they have the advantage that they are computationally inexpensive and provide the user with a tractable conceptual description of earthquake physics and the importance of the friction parameter. In our case, we were interested in investigating the effect of frictional parameter bias on the estimated fault states in earthquake cycle models. Hence in the present work, we investigated a simplified version of a Burridge-Knopoff spring-block slide model in a simplified 0D form. Eventually, to accurately simulate the behavior of real earthquake faults, 1D, 2D and 3D simulation models will be required (e.g., Li et al., 2021).*

Comment 5:
I suggest different colors for the trajectories in the phase diagram in Figure 2, as it is hard to distinguish between them.

Response:
This comment has been incorporated and the color has been changed in the figure.

Comment 6:
Comment 6:
4 time steps in this model correspond to which portion of a seismic event? Please, describe it in the manuscript to situate the reader on the frequency of the assimilation steps, as earthquake cycles may not be a subject well understood by many.

Response:
Having large assimilation steps can also have a detrimental effect on the data assimilation process as it can miss characteristic variations of the earthquake cycle. A parameter bias can have a substantial effect on the evolution of the state variables, which may be difficult to correct if the assimilation step is large. Hence a short time step is to be chosen that allows the assimilation to capture the important characteristics of the earthquake cycle. An additional text has been added in the manuscript (line 207-210, page 9) to educate the readers behind selecting this time step as follows:

*Large assimilation steps can adversely affect data assimilation, as they can miss variations in earthquake evolution. Thus, it is necessary to choose a short time step. In the present study, observations were sampled at 4 time units with the standard deviation of observation error as σβ is 0.6 for fault shear stress and 1.15 for slip velocity observations.*

Comment 7:
Have you tested the impact of the use of less particles in the filter? If so, it would be good to share these results as well.

Response:
The authors are thankful to the reviewer for the comment. Yes, the authors have used 50 and 100 particles for this study. However, using a smaller number of particles, the system faces filter degeneracy issues. The authors added a sentence to describe these results. We have addressed this comment by adding the following text in the manuscript (line 340-347, page 21) as:

*Another point of attention is the selection of the number of particles required for a correct sampling of the prior. On the one hand, increasing the number of particles can improve estimation accuracy but limited computational resources can make this impractical. On the other hand, having a lower number of particles increases the risk of filter degeneracy. In this study, we initially used 50 and 100 particles, but we increased the number of particles to 1000 to avoid degeneracy. Realization of 1000 particles may be computationally expensive for models that are more complex than the model used here. Improved sampling techniques, for example using a proposal density function, or using a particle flow (e.g. Hu and Van Leeuwen, 2021) could help to reduce computational costs while maintaining the advantage of a nonlinear filtering method.*

Comment 8:
How do the orders of magnitude of the observation errors compare to the states? Are those the typical magnitude of the real measurement errors?

Response:
The observational errors of the synthetic observations used in the study have been mentioned in the text (line 215-216, page 10). To obtain error estimates for each measurement type, the method explained in the study by Van Dinther (2019) were used where latest state-of-the-art

values from the literature were considered. These errors were then downscaled to our model setup using the analogue scaling relation developed in Corbi et al.2013 (Van Dinther (2019)).

Comment 9:
It would be good to explain in the label of Figure 5 what each of the lines in the pdf represent.

The red line represents the posterior pdf, the grey line represents the prior pdf, the magenta line shows where the observation stands, and the black line is the true state. This explanation has been added in the manuscript.

Comment 10:
The authors mention that by using different assimilation settings, it is possible to inflate the ensemble. But it seems that the ensemble spread is not exactly the problem, as the state-parameter estimation presents much better results than the other tests which have an improved ensemble spread. Can the authors explain more clearly the effects of the spread on this specific seismic system?

Response:
The authors are thankful to the reviewer for the comment. In the case of state parameter estimation, the prior distribution contains particles with different parameter values (covering the entire spectrum from $\varepsilon = 0.1$ to $0.8$) as shown in Figure 9 (a-e). Hence for the data assimilation to be effective there is no need to inflate the ensemble spread to cover the (observation or) the true state. As seen from Fig 9, the range of the prior encompasses the shear stress value of the observation. On the other hand, in state estimation, the parameter $\varepsilon$ is constant, which would require inflation of the ensemble to encompass the value of the observation.

Comment 11:
Figure 8b): What happens after nearly every 250 time steps, in which the periodic behaviour is lost by a double peak? What is the influence of the double resampling in these patterns?

Response:
In Figure 8b, which represents the double resampling experiment, we observe a double peak at 250 time steps. The data assimilation analysis does not fit the observations well. A similar mismatch is observed after approximately 500 time steps in this experiment. We thank the reviewer for pointing this out. At these moments, the double resampling effectively increases the spread in the particles to such an extent, that the constraint to the shear stress observations becomes less strong. The double resampling is not as effective in increasing the ensemble spread as the increased model error is (Fig 8a). We added a sentence to the text to describe this. We have addressed this comment by adding text as follows in the manuscript (line 269-274, page 17):

*It is also important to highlight that in Fig.8b, which represents the double resampling experiment, we observe a double peak at 250 time steps. At this time step, the data assimilation analysis does not fit the observations well. A similar mismatch is observed after approximately 500 time steps in this experiment. At these moments, the double resampling effectively increases the spread in the particles to such an extent, that the constraint to the shear stress observations becomes less strong. Hence, we can conclude that though*

*double resampling can be useful in retaining important particles in the prior distribution, it is not as effective in increasing the ensemble spread as the increased model error (Fig.8a).*

Comment 12:
Still on Figure 8, it seems that Fig 8a) presents better results than Fig 8c), which makes me wonder if the double resampling is really helping the systeI.

Response:
Indeed, the increased model error in Fig 8a appears to be more effective than the double resampling. We changed the text to reflect this and thank the reviewer for this observation. We have addressed this comment by adding text (line 274-279, page 17) in the manuscript.

Comment 13:
The results for the state-parameter estimation are indeed promising and I congratulate the authors for this, but I would expect a comparison of these results with any other study (if they exist) using 1D models with or without data assimilation. The manuscript lacks information on other results found by studies which used other data assimilation methods and/or models with different dimensions.

Response

Data assimilation in earthquake cycle models is still in its infancy. We are not aware of other studies that considers a model similar to the earthquake cycle model that we have used in this study. In the discussion, we have added a section which talks about the use of simplified and higher dimensional models for studying earthquakes. (line 352-362).

References:

Corbi, F., Funiciello, F., Moroni, M., Van Dinther, Y., Mai, P. M., Dalguer, L. A., & Faccenna, C. (2013). The seismic cycle at subduction thrusts: 1. Insights from laboratory models. *Journal of Geophysical Research: Solid Earth*, *118*(4), 1483-1501.

van Dinther, Y., Künsch, H. R., & Fichtner, A. (2019). Ensemble data assimilation for earthquake sequences: probabilistic estimation and forecasting of fault stresses. *Geophysical Journal International*, *217*(3), 1453-1478.